# Profound Impact of Local Climatic Conditions on IgE Sensitization Profiles: Evidence from Argentine Cities

**DOI:** 10.3390/ijms262412101

**Published:** 2025-12-16

**Authors:** Eszter Sarzsinszky, Paola Smaldini, Marcela Chinigo, María Ardanaz, Pablo Benítez, Ana Ramos, María Eugenia Braviz Lopez, Gonzalo Ramón, Germán Ramón, Thomas Schlederer, Mikhail Tulaev, Rudolf Valenta, Huey-Jy Huang, Susanne Vrtala, Guillermo Docena

**Affiliations:** 1Division of Immunopathology, Department of Pathophysiology and Allergy Research, Medical University of Vienna, 1090 Vienna, Austria; eszter.sarzsinszky@meduniwien.ac.at (E.S.); thomas.schlederer@meduniwien.ac.at (T.S.); mikhail.tulaev@meduniwien.ac.at (M.T.); rudolf.valenta@meduniwien.ac.at (R.V.); huey-jy.huang@meduniwien.ac.at (H.-J.H.); susanne.vrtala@meduniwien.ac.at (S.V.); 2Instituto de Estudios Inmunológicos y Fisiopatológicos-IIFP, Facultad de Ciencias Exactas, Universidad Nacional de La Plata, Consejo Nacional de Investigaciones Científicas y Técnicas (CONICET), La Plata 1900, Argentina; paolasmaldini@gmail.com; 3Hospital San Juan de Dios, La Plata 1900, Argentina; marcelapiachinigo@hotmail.com (M.C.); mfardanaz@hotmail.com (M.A.); pablonorbertobenitez@gmail.com (P.B.); ramosanaines@gmail.com (A.R.); labcsjdedios@gmail.com (M.E.B.L.); 4Instituto de Alergia e Inmunología del Sur, Bahía Blanca 8000, Argentina; gonza.ramon@hotmail.com (G.R.); germanramon2004@hotmail.com (G.R.); 5Laboratory of Immunopathology, Department of Clinical Immunology and Allergy, Sechenov First Moscow State Medical University, 119991 Moscow, Russia; 6Center for Molecular Allergology, Karl Landsteiner University for Healthcare Sciences, 3500 Krems, Austria; 7Life Improvement by Future Technologies (LIFT) Center, 121205 Moscow, Russia

**Keywords:** microarray, allergen, allergy, IgE, sensitization, Argentina, house dust mite, pollen, allergic diseases, molecular allergy diagnosis, climate

## Abstract

Allergen sensitization profiles are increasingly affected by environmental and climate changes. This study exemplifies fundamental differences in molecular IgE sensitization profiles in two nearby regions in Argentina with different climatic conditions (La Plata and Bahía Blanca). A cross-sectional study was conducted involving 155 patients with allergic symptoms from La Plata and Bahía Blanca (34.0 ± 11.2 years, female/male: 83/72). Serum samples were analyzed for IgE reactivity using a chip containing 101 micro-arrayed allergen molecules. Statistical analyses were performed to compare allergen-specific IgE levels, sensitization prevalences and reported symptoms. Patients from La Plata—with subtropical weather—showed a higher prevalence of IgE reactivity to house dust mite (HDM) allergens (Der p 23: 74%; Der p 1: 53% and Der p 2: 56%) and more frequently reported asthma (AS) symptoms (40% vs. 24%) than patients from Bahía Blanca. In contrast, patients from Bahía Blanca, with dry and windy weather, exhibited higher sensitization rates to pollen allergens, particularly Phl p 1 (49%) and Ole e 1 (22%) as well as to *Alternaria alternata* (Alt a 1, 35%) and reported a significantly higher prevalence of skin manifestations (54% vs. 31%) than those from La Plata. Cat allergen Fel d 1 was an equally important sensitizer in both regions (La Plata 30% and Bahía Blanca 37%). Sensitization to class 1 food allergens was rare in both groups (1–8%), including non-specific lipid transfer proteins (peanut Ara h 9 and peach Pru p 3) but IgE sensitizations to genuine peanut allergens were almost absent. Important regional differences in allergen sensitization profiles were observed between two geographically close regions with different climatic conditions. Our findings underscore the relevance of region-specific allergen profiling and highlight the clinical utility of molecular allergy diagnosis for a more precise allergen identification and improved management of allergic diseases.

## 1. Introduction

The global increase in allergic diseases is a major public health concern, with growing evidence that environmental changes play a key role. The rise in atmospheric CO_2_ levels has led to higher pollen yields, extended pollen seasons, and the expansion of fungi and HDMs due to increased surface humidity [1,2,3,4,5]. These changes disproportionally affect regions in the tropics and subtropics, including Argentina, which exhibits considerable climatic diversity. According to the Köppen–Geiger system classification, Argentina comprises eleven different climate types and six different climatic regions [6]. This study focuses on two major cities, La Plata and Bahía Blanca, which lie only 570 km apart, but in different climatic zones, in close proximity to the South Atlantic Ocean. La Plata is located in the humid pampas characterized by temperate weather and the absence of a dry season and hot summers (Cfa), whereas Bahía Blanca lies in dry pampas, with arid steppe conditions, windy, dry weather and colder average temperatures (BSk) (Figure 1).

Despite the known burden of allergic diseases, such as asthma and rhinoconjunctivitis in Argentina, no studies to date have used molecular allergy diagnosis [7] to analyze allergen-specific IgE sensitization in this region or even in Latin America [8,9,10]. Previous studies on allergic sensitization in Argentina and Latin America have relied primarily on allergen extract-based assays [11,12]. However, accurate identification of disease-eliciting allergen molecules is necessary for allergen-specific management of allergies, such as allergen-specific immunotherapy (AIT) or allergen avoidance [13].

This study is the first to apply an in-house-developed allergen microarray to assess molecular IgE sensitization patterns in adult allergic patients from two cities in Argentina, linking allergen-specific IgE profiles to clinical symptoms. We provide detailed insights into the regional variations in molecular sensitization to respiratory and food allergens, which are critical for improving the diagnosis and management of allergic diseases in Argentina. Importantly, our study reveals profound differences in two nearby areas due to regional climatic differences.

## 2. Results

### 2.1. Demographic and Clinical Characterization of the Study Population

The study included adolescents and adults with allergic symptoms, aged 15–62 years (mean 34.0 years). Serum samples were collected from patients attending allergy centers in La Plata (*n* = 55) and Bahía Blanca (*n* = 100) (Figure 1, Appendix A). The gender distribution was comparable between regions, with a slight predominance of females (82 females and 73 males overall). Bahía Blanca is situated at a latitude of −38.71° in an arid steppe (Bsk) climatic zone, whereas La Plata, located further north at a latitude of −34.92°, belongs to the humid subtropical (Cfa) climatic region (Figure 1).

Participants completed the International Study of Asthma and Allergies in Childhood (ISAAC) questionnaire, which provided data on allergic symptoms. The most frequently reported symptom was rhinoconjunctivitis (RC), with similar frequencies in both regions (93.0% in Bahía Blanca and 90.9% in La Plata). However, skin manifestations (D) were more frequent in Bahía Blanca (54.0%) compared to La Plata (30.9%). Conversely, asthma (AS) was more prevalent in La Plata (40.0%) than in Bahía Blanca (24.0%) (Figure 1).

Sera from all participants were analyzed for allergen-specific IgE reactivity using an in-house developed allergen microarray (Appendix A). Overall, 78.0% of participants (121/155) exhibited detectable IgE levels (cut-off ISU ≥ 0.1) to one or more allergens on the microarray (Figure 1). The number of allergens recognized per patient varied between regions, though differences were not statistically significant. In Bahía Blanca, 55.1% of sensitized patients recognized up to five respiratory allergens, 33.3% between six and ten, and 11.5% more than ten allergens. In La Plata, 51.2% of sensitized patients reacted to up to five respiratory allergens, 23.3% between six and ten, and 25.6% to more than ten allergens. These findings indicate a broader respiratory allergen sensitization profile in patients from La Plata (Appendix A).

### 2.2. Bahía Blanca and La Plata Show Different Molecular IgE Sensitization Patterns Associated with Different Climatic and Environmental Factors

IgE sensitization patterns of Bahía Blanca and La Plata revealed substantial differences, reflecting the distinct climatic and environmental conditions of the two regions. In La Plata, the most common IgE-reactive respiratory allergens were derived from HDM, with Der p 23 being the most frequently recognized allergen (74%), followed by Der p 2 (56%), Der p 1 (53%), Der p 7 (40%), Der p 21 (37%), Der p 5 (35%) and Blo t 21 (28%). Sensitization to pollen allergens, such as Phl p 1, was less common, namely 19% (Appendix A). The most frequently recognized animal-derived allergen was cat major allergen Fel d 1, with a recognition frequency of 30%, whereas IgE sensitization to dog allergen molecules was quite rare in both regions (Figure 2 and Appendix A).

In contrast, Bahía Blanca exhibited a higher prevalence of sensitization to pollen allergens, with timothy grass allergen Phl p 1 being the most frequent sensitizer (49%), followed by the olive pollen allergen Ole e 1 (28%), other grass pollen allergens (Phl p 2, 3 and 5) and the major mugwort allergen, Art v 1 (15%) (Figure 3 and Appendix A). The sensitization rate to the major cat allergen, Fel d 1 (37%), in Bahía Blanca was comparable to that in La Plata (Figure 2 and Figure 3 and Appendix A). Notably, IgE sensitization of the major *Alternaria* allergen Alt a 1 (35%) was more than twice as frequent in Bahía Blanca as in La Plata (Figure 2 and Figure 3 and Appendix A). Conversely, IgE sensitization to HDM allergens, including Der p 23 (33%) and Der p 2 (33%), was less than half that in La Plata, with Blo t 21 showing minimal prevalence (1%). Sensitization to Ole e 1 (28%) was substantially higher in Bahía Blanca than in La Plata (5%) (Figure 2 and Figure 3 and Appendix A).

These patterns correlate with regional climatic and land cover data obtained in the region (Appendix A). La Plata, with higher annual relative humidity (78.9%) and precipitation (1072.7 mm), provides favorable conditions for HDM proliferation, consistent with dominant HDM sensitization. Conversely, Bahía Blanca, characterized by lower annual precipitation (639.1 mm), lower relative humidity (64.6%), and higher wind flow speed (Bahía Blanca 18.8 km/h vs. La Plata 8.2 km/h), favors grass pollen and mold spore dispersal and persistence [14] consistent with the predominance of Phl p 1 and Alt a 1 sensitization. Furthermore, the larger proportion of pastureland in Bahía Blanca (64.3% of total area versus 25.4% in La Plata) likely increases the grass pollen exposure in Bahía Blanca, contributing to the observed sensitization differences. The higher frequency of Ole e 1 sensitization in Bahía Blanca is consistent with the presence of olive trees both in agricultural production areas and in urban spaces as ornamental trees.

### 2.3. Cross-Reactive Arginine Kinase and Tropomyosin Represent the Most Frequently Recognized Potential Food Allergens in Both Cities

Sensitization rates to food allergens were similar in both cities, ranging from 1% to 8%. In Bahía Blanca, the most frequently recognized potential food allergens were cross-reactive tropomyosins (TMs). Sensitization rates and IgE levels (median and maximum ISU) for food-derived TM were Pen m 1 from shrimp, 5% (2.4 and 91.0 ISU), Hel as 1 from snail, 6% (0.8 and 46.4 ISU); and Ani s 3 from Anisakis worm, 8% (0.3 and 39.1 ISU). For respiratory TM, the corresponding figures were: Der p 10, 5% (8.0 and 135.7 ISU), Bla g 7, 5% (7.6 and 122.1 ISU); and Blo t 10, 4% (3.7 and 53.6 ISU). Notably, all patients sensitized to Pen m 1 were co-sensitized to Der p 10. These findings suggest that the frequency and levels of specific IgE to food-derived TM were similar to those for the highly cross–reactive HDM tropomyosin, Der p 10, making it difficult to determine whether sensitization was primarily driven by food allergens, Der p 10, or other respiratory TM allergens (Figure 2 and Figure 3 and Appendix A).

IgE sensitization to plant-derived non-specific lipid transfer proteins (nsLTP), Ara h 9 (peanut, 2–6%) and Pru p 3 (peach, 2–5%) was found in both cities. For class 1 food allergens, sensitization to egg (Gal d 1) was 2–3%, slightly higher than milk sensitization (Bos d LF, 1–2%), consistent with previous evidence that egg allergy can persist into adulthood [15,16]. Sensitization to genuine peanut storage proteins (Ara h 1, Ara h 2 and Ara h 6) was rare (≤2%). No IgE sensitization to fish or wheat allergens was detected and milk sensitization was very rare in both cities.

IgE sensitization to pollen-related, cross-reactive class 2 food allergens was detected at low frequency. Bet v 1 homologues, including Gly m 4 (soy), Ara h 8 (peanut), Cor a 1 (hazelnut), and Pru p 1 (peach), were identified in a few patients (≤2–3%), consistent with secondary food allergy mediated by cross-reactivity to the major birch pollen allergen Bet v 1. Profilin-related food sensitizations were not directly assessed; however, based on 5–14% sensitized to the pollen profilins (Amb a 8, Bet v 2, Ole e 2, and Phl p 12), a similar rate of food profilin sensitization can be expected (Figure 2 and Figure 3 and Appendix A).

### 2.4. Peculiarities of IgE Sensitizations to Other Antigens

IgE sensitization to Ves v 5 from wasp venom was more frequent than to bee venom allergens and was more prevalent in La Plata (12%) compared with Bahía Blanca (9%). This finding is in line with previous reports showing that wasp sensitization is generally more common than bee sensitization in many populations [17,18,19]. Latex allergen Hev b 8 was detected in both regions, though at low prevalence (5% in La Plata and 1% in Bahía Blanca), and was mainly attributed to cross-reactive pollen profilins rather than genuine latex sensitization (Figure 2 and Figure 3 and Appendix A).

IgE sensitization prevalence to cross-reactive carbohydrate determinants (CCDs)- represented by HRP on the microarray-was comparable in patients of both cities (18.6% La Plata, median ISU 0.26; 17.9% Bahía Blanca, median ISU 0.17), but specific IgE levels were low (Figure 4, Appendix A).

### 2.5. Regional Variation in Allergic Symptoms in La Plata and Bahía Blanca

The prevalence of allergic symptoms differed significantly between La Plata and Bahía Blanca, reflecting regional variations in clinical presentations among study participants. Rhinoconjunctivitis was the most frequently reported symptom in both cities, affecting 91% of participants in La Plata and 93% in Bahía Blanca, with no significant difference observed between the regions (χ^2^ = 0.64, *p* = n.s.). La Plata, characterized by a humid subtropical climate (Cfa), had significantly higher asthma prevalence (40%), compared to 24% in Bahía Blanca (χ^2^ = 3.70, *p* = 0.04). In contrast, in Bahía Blanca, characterized by an arid steppe climate (Bsk), with a colder and windier winter, skin manifestations were more prevalent (54%) compared to La Plata (31%) (χ^2^ = 7.74, *p* < 0.01) (Figure 1 and Figure 5). No symptoms of food allergy were reported in the patients investigated in both cities.

### 2.6. High Sensitization Rates to Grass Pollen Allergen Phl p 1 and Fungal Spore Allergen from Alternaria alternata Alt a 1, with Evidence of Seasonal Overlap

Sensitization to *Alternaria alternata* (Alt a 1) and genuine grass pollen marker allergens (Phl p 1, 2 and 5) was highly frequent in both La Plata and Bahía Blanca, underscoring their clinical relevance in these regions. Of note, the combined use of Phl p 1, Phl p 2, and Phl p 5 identified the same proportion of grass pollen allergic patients as the broader panel including Phl p 1, 2, 3, 5, and 6. This indicates that these three genuine marker allergens are sufficient to identify all grass pollen allergic patients in our cohorts. In La Plata, 16.2% of participants were sensitized to Alt a 1 but not to grass pollen marker allergens, while 16.2% were sensitized to grass pollen marker allergens but not to Alt a 1 (Figure 4). Only 2.3% of participants showed co-sensitization to grass pollen and *Alternaria*. In Bahía Blanca, the prevalence of exclusive sensitization was higher for grass pollen allergens (34.6%) compared to *Alternaria* (16.6%), with a co-sensitization rate of 17.9%.

In 2005, the grass pollen season for *Poaceae* species extended from August to April, with peak concentrations observed in November (1706 grains/m^3^) (Appendix A). By comparison, *Alternaria* spores were present throughout the year, with peak concentrations during summer and autumn. Monthly spore counts ranged from 9000 spores/m^3^ in May to 14,500 spores/m^3^ in February, Ref. [20] indicating a consistent exposure burden across seasons (Appendix A). Together, these findings point to periods of overlapping exposure to grass pollen and *Alternaria* spores.

## 3. Discussion

Our study provides the first detailed insight into molecular IgE-sensitization profiles in Latin America, specifically in Argentina, using micro-arrayed allergen molecules in a cross-sectional study, revealing significant regional differences between La Plata and Bahía Blanca. Our findings underscore the influence of climatic conditions on allergic sensitization patterns, highlighting the different importance of house dust mite and grass pollen allergens in two cities of Argentina, La Plata and Bahía Blanca, respectively.

Interestingly, differences were observed in the IgE recognition profile and in the reported symptoms of the two regions analyzed. In La Plata, we identified major HDM allergens (Der p 23, 2, and 1) and cat allergen Fel d 1 as the most important indoor sensitizers, and patients from La Plata more often reported symptoms of asthma than patients from Bahía Blanca. This is in line with the observations that high frequencies of IgE sensitizations to HDM allergens significantly increase bronchial responsiveness in sensitized individuals [3,21,22]. In contrast, in Bahía Blanca, characterized by a drier and windier climate, compared to La Plata, patients with asthma were frequently sensitized to the fungal allergen Alt a 1, in line with findings by Peat JK et al. linking *Alternaria* exposure to asthma in dry rural regions [23,24,25]. In the study by Kiewet MBG et al., similar climate-related regional differences have been reported in Southern Europe (Spain), where Der p 23 prevalence was higher in subtropical areas [19]. However, it was found that Der p 2 typically dominates HDM sensitization [19]. This aligns with studies indicating Der p 23’s superior prevalence in subtropical regions, suggesting a climatic influence on allergen distribution and sensitization patterns [26,27].

Moreover, La Plata exhibited a high sensitization frequency to major allergens Blo t 5 (23%) and Blo t 21 (33%) from *Blomia tropicalis*, the mite prevalent in the tropics and subtropics [28,29], while patients in Bahía Blanca showed very low IgE recognition prevalence to these allergens. Sensitization to *Blomia* allergens might be explained by cross-reactivity to Der p allergens from the same allergen group; however, it is tempting to speculate that *Blomia tropicalis* represents an individual allergen source and must be distinguished from *Dermatophagoides pteronyssinus* in certain regions [28].

Conversely, Bahía Blanca exhibited higher sensitization rates to pollen allergens, particularly to the timothy grass allergen, Phl p 1, and to the fungal spore allergen Alt a 1.

Sensitization to Fel d 1, the major cat allergen, was very common in both regions (La Plata 30%, Bahía Blanca 37%), which is not surprising given Argentina’s high rate of pet ownership and its high cat population of approximately 3 million cats [30,31].

Interestingly, IgE sensitization to dog allergens, Can f 1, Can f 2, and Can f 4 was much less prevalent. This may be due to the fact that dogs are kept more often outside than cats but one must also consider that we have not included the major dog allergen Can f 5 in the panel of micro-arrayed allergens tested and thus did not identify all dog-sensitized subjects. Among tree pollen allergens, Ole e 1 was the most prominent, with a prevalence of 5% in La Plata and 28% in Bahía Blanca. Sensitization to Ole e 1 may be attributable to the presence of olive trees in both rural and urban areas; however, cross-reactivity of IgE antibodies specific to ash pollen allergen Fra e 1 [32], recognized as the second most abundant pollen-producing tree in Bahía Blanca [33], should also be considered. Nevertheless, given Argentina’s status as a leading producer and exporter of olive oil in the Americas, with an estimated 90,000 hectares of olive cultivation, including significant areas in Buenos Aires province, genuine sensitization to Ole e 1 remains highly plausible, as *Olea* pollen grains have been detected in Bahía Blanca in the air (Appendix A). It is a limitation of our study that certain tree pollen allergens are underrepresented on our microarray, potentially missing other species relevant to Argentina’s diverse tree population. Studies indicate that *Cupressaceae* pollen is an important pollen allergen source in Bahía Blanca [33,34] and cypress is a major contributor to respiratory allergies in Mediterranean climates, which share similarities with parts of Argentina [35]. However, the major cypress allergens are glycosylated proteins and the expression of functional IgE-reactive major allergens devoid of IgE-reactive CCDs has not been solved yet.

In contrast to regions like Northern Europe, where Bet v 1 (birch pollen allergen) dominates the hierarchy of aeroallergen sensitization [19,36,37], our study revealed very low recognition frequencies for this allergen in Argentina. Although birch trees have been introduced to Argentina, they are not native to the area, and the proportion of birch trees compared to native species is minimal, which explains the low prevalence of PR-10 sensitization in the population. Bet v 1 was recognized only by 5% of the study population, and the IgE recognition of cross-reactive PR10 class 2 allergens such as Cor a 1.041, Gly m 4, Ara h 8, Mal d 1 and Pru p 1 was even lower.

Studies from other subtropical regions like South Africa show frequent sensitization to the egg allergen Gal d 1 among food allergens in children [38]. Gal d 1 was also one of the most frequently detected class 1 allergens besides nsLTPs from peanut (Ara h 9) and peach (Pru p 3) in patients from La Plata and Bahía Blanca. IgE sensitization to shellfish allergens, tropomyosin and arginine kinase was frequent, which aligns with numerous publications that identify shellfish allergy as the most prevalent food allergy among adults in the United States [39,40], Asia [41,42] and other regions [43,44,45]. Moreover, shellfish are a leading cause of food-induced anaphylaxis worldwide [46,47,48,49]. However, it is quite possible that IgE reactivity to the latter allergens was due to IgE cross-reactivity with the homologous allergens from HDM. Of note, IgE sensitization to Ves v 5 was very common, affecting between 9 and 12% of the patients, whereas IgE sensitization to bee venom allergens was rare. Like in Europe, Ves v 5 is therefore the most frequently recognized venom allergen in our population.

Out of 155 patients reporting allergic symptoms, 78% showed IgE sensitization to allergens included on our microarray. This may be due to the fact that certain allergens, such as Can f 5 and cypress allergens, were not included in our micro-array, but it is even more likely that the ISAAC questionnaire may frequently yield positive results for subjects without IgE sensitizations. In fact, the presence of symptoms such as rhinitis, asthma or dermatitis does not necessarily indicate IgE-mediated sensitization. Respiratory virus infections, in particular rhinovirus, are well-established triggers of wheeze, asthma, and rhinitis, and can exacerbate allergic symptoms or mimic them in non-sensitized individuals. Recent studies have highlighted that virus-driven airway inflammation can account for acute exacerbations both in IgE-sensitized and non-sensitized patients, supporting the notion that symptomatic but IgE-negative cases may represent virus-induced disease rather than allergy [50,51].

Our study also highlights another important clinical problem that can be addressed by molecular allergy diagnosis. Allergen extract-based assays, which are still widely utilized by practitioners, can yield false-negative and positive results due to the absence of key allergens or cross-reactivity caused by allergen glycosylation and IgE-reactive CCDs, leading to potential misdiagnoses and suboptimal treatment strategies. We found considerable sensitization rates to CCDs, major grass pollen and *Alternaria alternata* allergens, which are present in the air simultaneously (Appendix A). This underlines the critical need for accurate differential diagnosis to identify the symptom-eliciting allergen and to guide targeted therapy such as allergen-specific immunotherapy. Indeed, grass pollen extracts often contain cross-reactive carbohydrate determinants [52,53] leading to false-positive results, while *Alternaria* extracts may lack Alt a 1 [54,55], the major allergen essential for diagnosing *Alternaria* allergy. Molecular allergy diagnosis, as demonstrated by our allergen microarray, provides a precise method for identifying the relevant allergen sources and overcomes the limitations of extract-based assays. We propose a clinically applicable diagnostic decision tree to translate molecular IgE results into therapeutic recommendations (Figure 4). Patients with allergy symptoms during the grass pollen and mold season should undergo IgE testing by molecular allergy diagnosis. Sensitization to the genuine grass pollen markers Phl p 1, 2, and 5 [56,57] confirms grass pollen allergy, while sensitization to the major *Alternaria* marker Alt a 1 [58] establishes *Alternaria* allergy. Patients positive for both sets of markers are classified as co-sensitized. In contrast, patients showing IgE reactivity restricted to minor grass allergens or only CCDs are classified as cross-reactive without clinical relevance. Based on this stratification, tailored allergen immunotherapy can be recommended: grass AIT for grass-sensitized patients, *Alternaria* AIT for Alt a 1-positive patients, combined AIT for co-sensitized patients, and no AIT in cases of cross-reactivity or negative results. This approach ensures that AIT is only offered when genuine sensitization is demonstrated, minimizing unnecessary treatment while addressing the true drivers of allergic disease.

The main limitation of this study is the small sample size in both cities. Another important limitation, considering Argentina’s diverse climatic conditions, is the exclusion of populations from the northern (BSh) and western (BWk) regions (Figure 1). Further research is warranted to assess sensitization patterns in these areas.

Argentina’s diverse climatic regions and geographical variations (e.g., Andes Mountains) might influence allergen profiles; therefore, future studies collecting samples from a broader area and larger cohorts will be necessary to comprehensively map allergen sensitization across the entire country. The ongoing assessment of IgE patterns and vegetation shifts due to climate change is critical, as it may predict emerging allergenic threats as climate zones shift towards temperate regions. Research studies based on tailored research microarrays will be crucial to precisely target symptom-eliciting allergens and improve immunotherapy for patients suffering from allergic diseases.

## 4. Materials and Methods

### 4.1. Cohorts and Design of the Study

The IgE sensitization profiles in two different regions in Argentina, Bahía Blanca and La Plata, were examined. A total of 155 serum samples (patients aged 15–62, mean age 34.0, 83 females and 72 males) were collected from adolescents and adults with allergic symptoms in allergy centers of La Plata (n = 55; 33.9 ± 12.1 years, female/male 40/15) and Bahía Blanca (n = 100; 34.1 ± 10.8 years, female/male 43/57) (Figure 1 and Appendix A). Patients were recruited from a public hospital in La Plata, Argentina (Hospital San Juan de Dios de La Plata) and two centers in Bahía Blanca, Argentina (Instituto de Alergia e Inmunología del Sur and Hospital Italiano Regional del Sur). Participants were randomly selected, regardless of symptoms or disease severity. No selection bias was introduced, and the study population is considered representative of the general adult population of both cities.

Our study aimed to compare the allergen sensitization patterns between these geographically and environmentally distinct areas (Figure 1). Participants completed an ISAAC questionnaire modified for adults to record symptoms indicative of allergy and disease severity [59]. Specific questions from the questionnaire addressing symptoms related to possible allergic rhinitis, skin manifestations, and asthma were summarized and selected for further analysis (Appendix A). Ethics approval was acquired and informed consent was obtained from all subjects involved in the study. The protocol and informed consent forms were analyzed and approved by the Ethics Committee of the Hospital San Juan de Dios de La Plata (LP3680/022021). Anonymized sera were analyzed in the Medical University of Vienna, Department of Pathophysiology and Allergy Research, with the permission of the ethics committee of the Medical University of Vienna, EK1641/2014. Participants aged 15–17 years were included with written parental consent, in accordance with the ethics approval.

### 4.2. Immunoglobulin E Measurements

For the evaluation of IgE sensitization, serum samples were analyzed using an allergen microarray chip produced by the Allergochip working group (see authors), which includes 101 different recombinant and natural allergens, including a cross-reactive carbohydrate determinant (CCD) marker (Appendix A). The protein printing procedure was performed as previously described [60]. Allergens were printed onto glass slides comprising six microarrays within an epoxy frame (Paul Marienfeld GmbH & Co. KG, Lauda-Königshofen, Germany). Slides were pre-coated with an amino-active polymer, MCP-2 (Lucidant Polymers, Sunnyvale, CA, USA), to facilitate protein immobilization. Allergens were spotted in triplicate in 75 mM Na_2_HPO_4_/NaH_2_PO_4_ buffer at pH 8.4 in a concentration ranging from 0.5 to 1 mg/mL using a SciFlexArrayer S12 (Scienion AG, Berlin, Germany). For allergen-specific IgE measurements, slides were washed with phosphate-buffered saline containing 0.5% Tween 20 (PBST) and dried by centrifugation. To each array, 30 µL of human serum was added along with controls, including sample diluent (SD) (Thermo Fisher, Waltham, MA, USA) and an in-house IgE calibration serum containing a mix of sensitized individuals’ sera. The slides were incubated for 2 h at room temperature. After washing, arrays were incubated with mouse anti-human IgE (Roche, Basel, Switzerland) conjugated with DyLight 550-2xPEG NHS Ester amine-reactive fluorescent dye (Pierce, Rockford, IL, USA) for 30 min in a final concentration of 1 μg/mL. Following a washing and drying step, slides were scanned using a confocal laser scanner (Tecan, Männedorf, Switzerland).

### 4.3. Data Analysis

Microarray image analysis was processed using MAPIX microarray image acquisition and analysis software Version 8.5.0 (Innopsys, Carbonne, France). The fluorescent intensities (FI) of IgE values were normalized against the sample diluent, and an IgE calibration serum was used to establish a linear conversion equation to ISAC standardized units (ISU). A percentage of patients with IgE sensitization were grouped according to their ISU class (low ≥ 0.1–1 ISU, moderate = 1–15 ISU, high > 15) (Figure 2 and Figure 3). Allergen molecules were categorized by their source (e.g., animal dander and food allergens) and sensitization route. Sensitization rates of patients from two different regions, La Plata (Appendix A) and Bahía Blanca (Appendix A), were calculated for each allergen, and allergens were ranked by their IgE recognition frequencies. Statistical analyses were performed to identify any significant regional differences in reported allergic symptoms and to evaluate differences in allergen-specific IgE levels and their correlation with reported symptoms. Chi-square (χ^2^) test was used to assess the statistical significance. All statistical analyses were conducted using GraphPad Prism 8 software.

### 4.4. Climatic and Geospatial Data Acquisition and Map Generation

Maps were generated using QGIS software (version 3.38.2; open-source Geographic Information System software) to visualize the spatial distribution of climate zones and calculate land cover in Argentina. Monthly climatological statistics, including average temperature, maximum and minimum temperatures, relative humidity, wind speed and precipitation for the period 1991–2020, were obtained from the official meteorological website of Argentina, Servicio Meteorológico Nacional (SMN). The Köppen–Geiger climate classification data by Beck et al. [6,61] covering the same period (1991–2020) were obtained in GeoTIFF format from https://www.gloh2o.org/koppen/, with a resolution of 1 km (accessed on 6 May 2025). Land cover information was provided by the MapBiomas Argentina Project, from “Collection 1” of the Annual Land Use and Land Cover Maps of Argentina, accessed in 2022. In QGIS, a base map of Argentina, displaying provincial boundaries and climate regions, was created by importing the vector and raster datasets. To refine the focus on this region, the raster climate data were clipped using the provincial boundary as a mask. An inset map was generated, magnifying the selected province while maintaining the context of the national map.

## Figures and Tables

**Figure 1 ijms-26-12101-f001:**
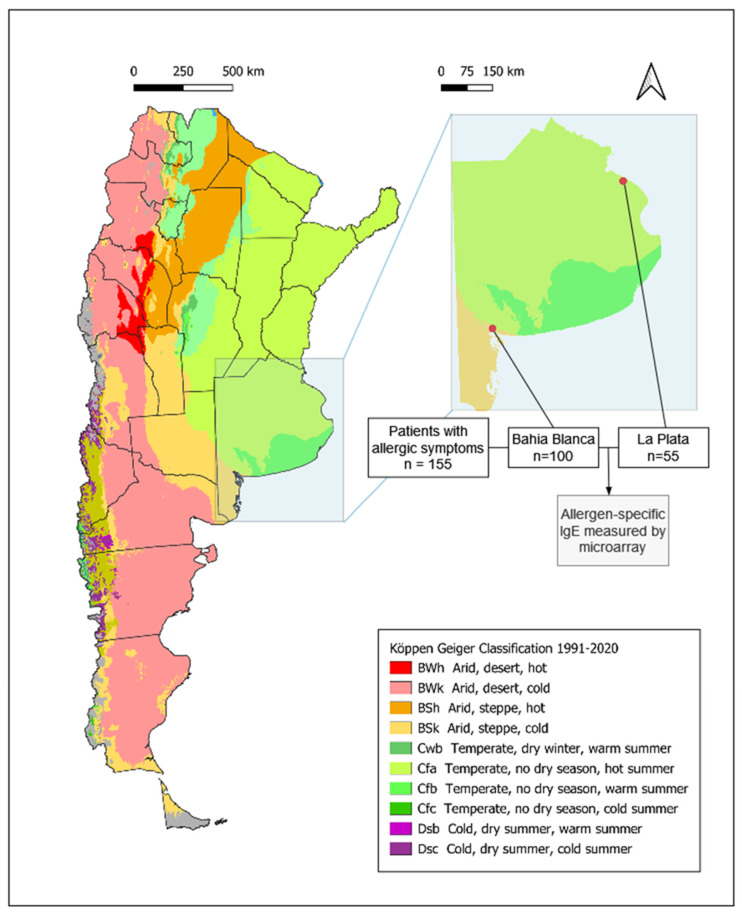
Flow chart illustrating the process of patient selection and involvement in the analyses for our study. Köpper–Geiger climate classification map for Argentina (1991–2020) modified according to Beck HE et al. [6] Demographic and clinical characterization of patients. Number and percentage of patients reporting symptoms of rhinoconjunctivitis, cutaneous manifestations and asthma within the specific regions of Bahía Blanca, La Plata, and the total study population. Symptoms were assessed using the ISAAC questionnaire.

**Figure 2 ijms-26-12101-f002:**
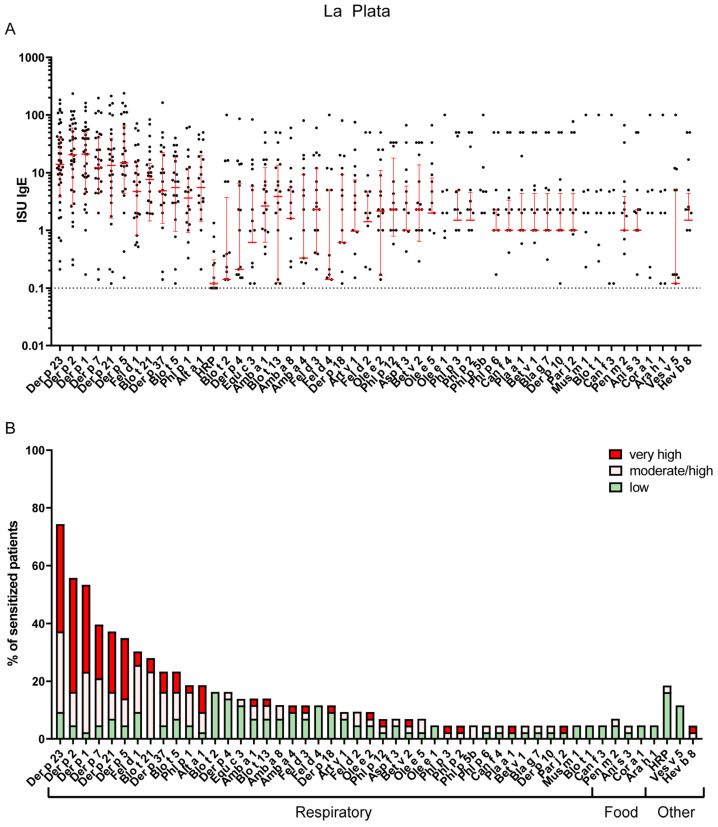
(**A**) IgE levels of the most frequent sensitizers in La Plata. Error bars (red) indicate medians ± interquartile ranges of positive ISU values. Cutoff for positive results (>=0.1) is indicated by a dashed horizontal line. Values below the cutoff are not shown. (**B**) Most frequently recognized respiratory, food and other allergens in La Plata (prevalence of IgE sensitizations of at least 5% is shown). The percentage of sensitized patients is divided into three different ISU gradings reflecting allergen-specific IgE levels: 0.1 < ISU < 1 low; 1 < ISU < 15 moderate/high; ISU ≥ 15 very high.

**Figure 3 ijms-26-12101-f003:**
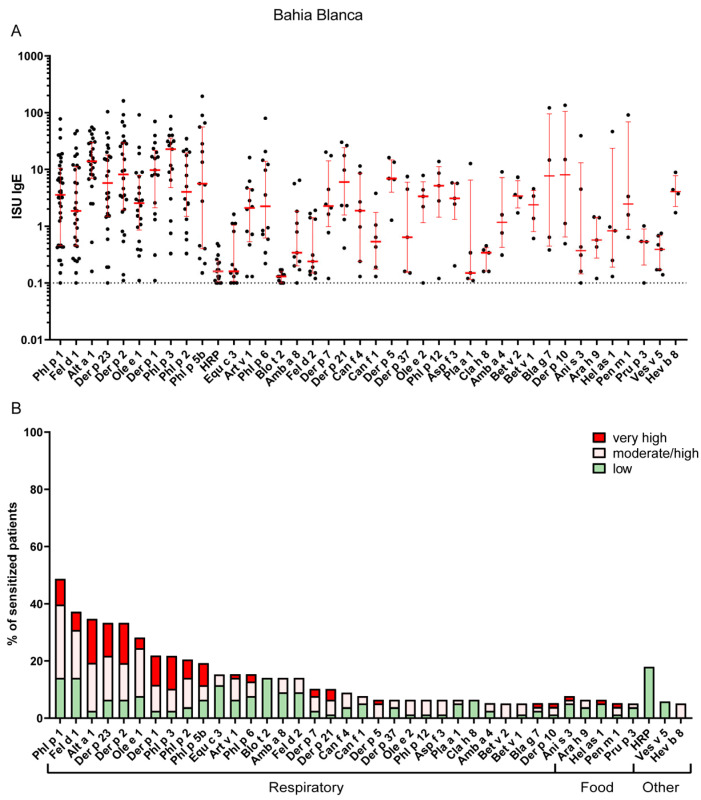
(**A**) IgE levels of the most frequent sensitizers in La Bahía Blanca. Error bars (red) indicate medians ± interquartile ranges of positive ISU values. Cutoff for positive results (>=0.1) is indicated by a dashed horizontal line. Values below the cutoff are not shown. (**B**) Most frequently recognized respiratory, food and other allergens in Bahía Blanca (prevalence of IgE sensitizations of at least 5% is shown). The percentage of sensitized patients is divided into three different ISU gradings reflecting allergen-specific IgE levels: 0.1 < ISU < 1 low; 1 < ISU < 15 moderate/high; ISU ≥ 15 very high.

**Figure 4 ijms-26-12101-f004:**
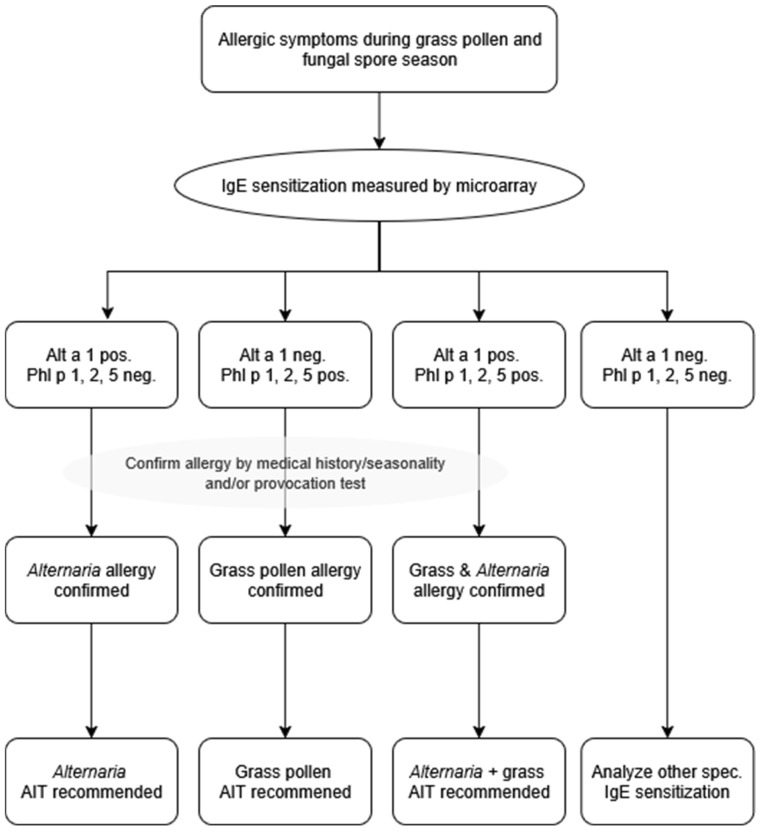
Numbers and percentages of patients sensitized to Alternaria alternata major allergen, Alt a 1 only (“Alt a 1 only”); or to any of the genuine grass pollen allergens, Phl p 1, 2 or 5 (“Grass only”) or to both (“Co-sensitization”), and to HRP, as a marker for cross-reactive carbohydrate determinants (CCD) recognition. Molecular allergy diagnosis to facilitate the correct prescription of allergen-specific immunotherapy for Alternaria and/or grass pollen allergy.

**Figure 5 ijms-26-12101-f005:**
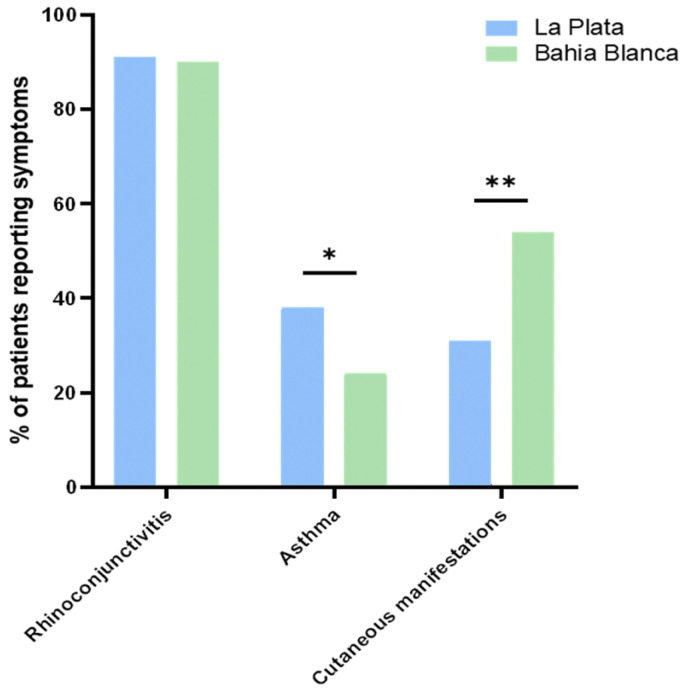
Percentage of patients reporting symptoms compared between Bahía Blanca and La Plata. Asterisks denote statistical significance at * *p* ≤ 0.05, and ** *p* ≤ 0.01.

## Data Availability

The original contributions presented in this study are included in the article/Appendix A. Further inquiries can be directed to the corresponding author.

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
