# Peer review of "Profound Impact of Local Climatic Conditions on IgE Sensitization Profiles: Evidence from Argentine Cities"

_ijms, 2025, doi:10.3390/ijms262412101_

Round 1
Reviewer 1 Report
Comments and Suggestions for Authors
This is a well written epidemiological study showing remarkable differences in the IgE molecular profiles of sensitizations of patients with allergic diseases examined in two Argentinian cities with environmental/climatic differences. Epidemiological studies of allergic diseases have been so far supported by sensitization tests based on allergen extracts. We need instead to move this field in the molecular allergology area. The study by Sarzsinszky et al. is therefore going into what I would consider "the right direction".
The greatest limitation of this study is the lack of information about inclusion criteria. In particular, we do not know how the subjects examined are representative of the general populaiton of the allergic patients in their respective cities. Therefore, the Authors cannot exclude that the observed differences may reflect differences in the characteristics of the study centers, of socioeconomic status of the population they are serving, selection or participation biases, etc. To improve their paper, the Authors should: (A) report in the materials and methods section many more details about the inclusion and exclusion criteria of the patients, and whether the IgE tests have been done as an add on to the clinical routine diagnostic work-up or they are performed in all the patients, independently from the disease category and severity; (B) include in the discussion a section on study limitations (I believe this section is missing) and list the limitations in the study design (especially if just consecutive patients have been examined).
Author Response
This is a well written epidemiological study showing remarkable differences in the IgE molecular profiles of sensitizations of patients with allergic diseases examined in two Argentinian cities with environmental/climatic differences. Epidemiological studies of allergic diseases have been so far supported by sensitization tests based on allergen extracts. We need instead to move this field in the molecular allergology area. The study by Sarzsinszky et al. is therefore going into what I would consider "the right direction".
-We thank the reviewer for their valuable comments.
The greatest limitation of this study is the lack of information about inclusion criteria. In particular, we do not know how the subjects examined are representative of the general populaiton of the allergic patients in their respective cities. Therefore, the Authors cannot exclude that the observed differences may reflect differences in the characteristics of the study centers, of socioeconomic status of the population they are serving, selection or participation biases, etc. To improve their paper, the Authors should: (A) report in the materials and methods section many more details about the inclusion and exclusion criteria of the patients, and whether the IgE tests have been done as an add on to the clinical routine diagnostic work-up or they are performed in all the patients, independently from the disease category and severity;
-We thank the reviewer for their insightful comments, which have contributed to improving the quality of the manuscript. Patients were recruited from a public general hospital of La Plata (Hospital San Juan de Dios de La Plata, Allergy Unit and Central Laboratory) and from two centers in Bahía Blanca: a private clinic (Instituto de Alergia e Inmunologia del Sur) and a hospital (Hospital Italiano Regional del Sur, Allergy Unit). Patients attended the different centers for the first consultations and were randomly selected for inclusion in the study, irrespective of symptoms or disease severity. The ISAAC questionnaire was administered to all patients.
No selection bias was introduced, and we consider that the study population is representative of the general adult population of both cities. The only exclusion criterion applied was age (participants were adults only). Molecular allergy diagnosis was not part of the routine clinical assessment; it was conducted exclusively for research purposes in all recruited participants, and all collected samples were analyzed.
This clarification has been incorporated into the Materials and Methods section in red, which now reads as follows on lines 368-373:
“The participants recruited were all attending their first consultation and were from a public hospital in La Plata (Hospital San Juan de Dios de La Plata) and two centers in Bahía Blanca (Instituto de Alergia e Inmunología del Sur) and Hospital Italiano Regional del Sur. Participants were randomly selected, regardless of symptoms or disease severity. No selection bias was introduced, and the study population is considered representative of the general adult population of both cities.”
(B) include in the discussion a section on study limitations (I believe this section is missing) and list the limitations in the study design (especially if just consecutive patients have been examined).
-As suggested by the reviewer a paragraph was added in the Discussion section in red to clarify this point. Now it reads as follows (L354-358):
“The main limitation of this study is the small sample size in both cities. Another important limitation, considering Argentina’s diverse climatic conditions, is the exclusion of populations from the northern (BSh) and western (BWk) regions (Figure 1). Further research is warranted to assess sensitization patterns in these areas.”

Reviewer 2 Report
Comments and Suggestions for Authors
- Evaluation of Allergen components’ sensitization is extremely powerful and useful tool for assessing the profile of sensitization of allergic patients. It gives opportunity for detailed diagnosis and personalized treatment. Studying the link between exposure and sensitization is as old as allergology itself. The allergy components provide new level of these investigations. The present study is a good example of this type of works. It provides very well depicted picture of the sensitization of population of two distinct regions of Argentina. It proves the idea that different climates predispose to different allergens’ profile and this in its turn predisposes to different population’s sensitization and possibly to different allergy symptoms. The strong side of the present work is the tailored list of allergen components, which proves the difference of sensitization of population in deferent climate areas. Unfortunately there is no data about the presence of allergens of question in the environment of the sensitized subjects.
- Manuscript provides thorough and up to date analysis of the problem, provides new data, which enriches the knowledge about the sensitization to various allergens in Argentina..
- I find the methodology of the manuscript good, as well as the presentation of articles reviewed.
- The conclusions are consisted with the arguments presented, they address the main question.
- Reference is appropriate and up to date.
- I found Tables 2 and 3 quite difficultly readable. Perhaps the labels on X-axis have to be at larger space or with smaller letters .
Author Response
Evaluation of Allergen components’ sensitization is extremely powerful and useful tool for assessing the profile of sensitization of allergic patients. It gives opportunity for detailed diagnosis and personalized treatment. Studying the link between exposure and sensitization is as old as allergology itself. The allergy components provide new level of these investigations. The present study is a good example of this type of works. It provides very well depicted picture of the sensitization of population of two distinct regions of Argentina. It proves the idea that different climates predispose to different allergens’ profile and this in its turn predisposes to different population’s sensitization and possibly to different allergy symptoms. The strong side of the present work is the tailored list of allergen components, which proves the difference of sensitization of population in deferent climate areas. Unfortunately there is no data about the presence of allergens of question in the environment of the sensitized subjects.
Manuscript provides thorough and up to date analysis of the problem, provides new data, which enriches the knowledge about the sensitization to various allergens in Argentina..
I find the methodology of the manuscript good, as well as the presentation of articles reviewed.
The conclusions are consisted with the arguments presented, they address the main question.
Reference is appropriate and up to date.
-We thank the reviewer for their valuable comments.
I found Tables 2 and 3 quite difficultly readable. Perhaps the labels on X-axis have to be at larger space or with smaller letters .
-We guess the reviewer referred to Figures 2 and 3. These figures have been modified.
